# An Optimized Method to Culture Human Primary Lung Tumor Cell Spheroids

**DOI:** 10.3390/cancers15235576

**Published:** 2023-11-25

**Authors:** Amanda Mueggler, Eléa Pilotto, Nadja Perriraz-Mayer, Sicong Jiang, Alfredo Addeo, Benoît Bédat, Wolfram Karenovics, Frédéric Triponez, Véronique Serre-Beinier

**Affiliations:** 1Division of Thoracic and Endocrine Surgery, Department of Surgery, Faculty of Medicine, Geneva University Hospitals, 1205 Geneva, Switzerlandnadja.mayer@unige.ch (N.P.-M.); sicong.jiang@etu.unige.ch (S.J.); benoit.bedat@hcuge.ch (B.B.); wolfram.karenovics@hcuge.ch (W.K.); frederic.triponez@hcuge.ch (F.T.); 2Department of Oncology, Geneva University Hospitals, 1205 Geneva, Switzerland; alfredo.addeo@hcuge.ch

**Keywords:** lung cancer, patient-derived spheroid, drug sensitivity, drug screening

## Abstract

**Simple Summary:**

Lung cancer, responsible for nearly 20% of global cancer-related deaths annually, is the leading cause of cancer mortality. Traditional treatments, like chemotherapy and radiation, are not always effective or appropriate due to the heterogeneity of lung cancer across patients. To address this challenge, our research introduces an approach to develop patient-derived spheroids (PDS) that are cultivated using cells from patient tumors and adjacent healthy tissue. These PDS allow us to mimic the unique microenvironment of patients, introducing a platform to assess personalized responses to drug treatments. We used this system to characterize patient cell populations, evaluate gene expression, and assess the sensitivity of PDS to specific drugs. By introducing a patient-specific platform to test drug sensitivity, our study holds the potential to enhance the efficacy of lung cancer treatments, paving the way for individualized and more effective lung cancer therapies in the future.

**Abstract:**

Lung cancer is the leading cause of cancer mortality worldwide, with a median survival rate at 5 years of less than 20%. While molecular mapping aids in selecting appropriate therapies, it cannot predict personalized treatment response and long-term efficacy. For addressing these challenges, there is a great need for functional tests. Within this context, we developed patient-derived spheroids (PDS) from tumor and adjacent normal tissue to biomimic the respective tissue for assessing the personalized drug treatment response in vitro. Surgically resected lung specimens were used to generate spheroids using a two-step culture procedure. Flow cytometry and immune staining enabled the characterization of different cell populations resulting from the lung samples. PDS phenotype, cell proliferation and apoptosis were evaluated. Differential gene expression between tumor and adjacent normal tissue was analyzed via RT-qPCR. PDS drug sensitivity was assessed using a cell metabolic assay in response to two chemotherapeutic drug combinations. Cellular and molecular analysis revealed the proportion of epithelial cells, fibroblasts, and immune cells in the patients’ tissue samples. Subsequently, PDS models from tumor and normal lung were successfully established using the expanded epithelial cells. As a proof of concept, an analysis of the drug treatment using PDS of lung adenoid cystic carcinoma exhibited a dose-dependent effect in response to cisplatin/etoposide and cisplatin/paclitaxel. Our spheroid model of both tumor and non-tumor lung cells holds great promise for enhancing the treatment efficacy in the cancer patients.

## 1. Background

Lung cancer is the leading cause of cancer death in adults [1]. In 2020, approximately 1.8 million people died of lung cancer worldwide, accounting for 18% of all cancer-related deaths. Between 80 and 85% of lung cancer cases are non-small-cell lung cancers (NSCLC), of which approximately 25–30% are primarily squamous cell carcinoma (SqCC), 40% are adenocarcinomas (Adc), and 10–15% are large-cell carcinoma (LCC) [2]. Although current cutting-edge targeted therapies have made progress in improving survival, the prognosis of lung cancer remains dismal. Currently, the 5-year survival rate for lung cancer is less than 20% and drops to as low as 6% for late-stage NSCLC [3].

Treatment options for lung cancer include surgery, chemotherapy, radiation therapy, targeted therapy, and immune checkpoint inhibitors (ICI). The choice of treatment depends on many factors, including staging, histology and genetic features, as well as the patient’s comorbidities and performance status. Chemotherapy is usually the primary treatment modality because most patients are ineligible for surgery. Although chemotherapy can extend survival, it is not curative in patients with advanced cancer, and many may not tolerate the side effects of the most potent regimens [4]. Lung cancer is also characterized by a high degree of heterogeneity, driving a need to tailor treatment to the different subtypes of the disease. In the last decade, concerted efforts to characterize the genomic landscape of lung cancer subtypes have made it possible to identify novel therapeutic targets and develop targeted therapies. For patients with advanced NSCLC for whom driver mutations are identified, targeted therapies like tyrosine kinase inhibitors (TKIs) have become the standard of care and have shown improved efficacy and lower side effects compared to chemotherapy [5]. Although targeted therapy can aid in disease control, NSCLC patients eventually develop resistance through a variety of mechanisms that result in cancer progression [6]. More recently, immunotherapy aimed at altering immune checkpoint inhibition through the cytotoxic T-lymphocyte-associated antigen 4 (CTLA-4) and mostly through programmed cell death protein (PD1/PD-L1) pathways [7,8,9,10] has become the standard of care for patients with NSCLC who do not have driver gene mutations. Therapies combining different chemotherapy treatments or combining immunotherapy with chemotherapy or targeted treatment have been the focus of extensive research and have already entered clinical practice with some promising preliminary results [11,12,13]. Despite this, due to the heterogeneous nature of lung cancer, it is challenging to predict whether an individual patient will respond to a given therapy.

Robust ex vivo models may thus represent promising tools to help identify and guide individualized treatment. These models should be established rapidly and deliver results within 2 to 3 weeks, which is clinically equivalent to the time between diagnosis and treatment decision. This short time frame highlights why animal models are inappropriate. In vitro three-dimensional (3D) tumor models, including organoids and spheroids, have proven to be relevant for the screening of new treatments [14,15]. Patient-derived organoids (PDOs) and patient-derived spheroids (PDS) have been successfully developed for various human cancers [16,17,18] and are emerging as promising new tools to guide clinically relevant individualized cancer therapy [16,19,20,21,22,23].

In this study, we established a simple lung PDS model that mimics the original tumor and used it to assess sensitivity to different chemotherapeutic treatments. We selected this ex vivo model because PDS are more amenable to high-throughput drug screening than PDO. Spheroids are also more reproducible in terms of shape and size compared to organoids, and are established without the need for exogenous extracellular matrix. Using a two-step method of culture, we successfully generated PDS from the most common subtypes of lung cancer and from their non-malignant adjacent tissues. The expression pattern of lung specific markers was maintained in most PDS, and genomic analysis indicated that 50% of the spheroids maintained a tumor phenotype, as previously described for lung organoids [18,24]. Finally, we successfully measured the sensitivity of adenoid cystic carcinoma PDS to two clinically relevant chemotherapy drug combinations. Our findings suggest that this lung PDS model is an effective ex vivo drug screening tool to reliably identify an optimized treatment that is tailored to the patient. This is the first step towards developing a more complex heterotypic PDS model that incorporates fibroblasts and immune cells to test the efficacy of all cancer treatments, including immunotherapy.

## 2. Materials and Methods

### 2.1. Reagents

All the reagents are listed in Appendix A.

### 2.2. Human Specimens and Adherent Cell Culture

This study was conducted according to the Declaration of Helsinki [25], and approved by the Swiss Ethics Committee on research involving humans (2018-02395, approved in 2018). Surgical specimens from untreated lung cancer were obtained from patients with informed consent from the thoracic and endocrine surgery division of the university hospitals of Geneva.

Surgically resected tissues were processed within 1 day of removal from patients. They were minced (approximately 1 mm^3^), washed and enzymatically digested for 1 h at 37 °C in AdDMEM/F12 medium (Advance Dulbecco’s modified Eagle’s medium/F12, PAN™Biotech GmbH, Aidenbach, Germany) supplemented with 0.3 mg/mL collagenase type I, 0.1 mg/mL collagenase type II (both from PAN™Biotech), 0.025 mg/mL elastase (Alfa Aesar, Haverhill, MA, USA), and 25 µg/mL DNAse (Roche, Basel, Switzerland), with gentle agitation [26]. Digested tissue suspensions were filtered through a 100-µm strainer and washed with AdDMEM/F12 medium. After red blood cell lysis in RBC lysis buffer (Biolegend, San Diego, CA, USA), the cells were cultured in complete culture medium.

### 2.3. Patient-Derived Spheroid Culture (PDS)

PDS composed of 1000 cells/spheroid were generated from 256-microwell agarose casts using the agarose 3D microwell technique. These 256-microwell agarose casts were generated following the manufacturer’s instructions. Briefly, 500 µL sterile agarose solution (Sigma-Aldrich, Livonia, MI, USA) at a 2.5% concentration, heated at 90 °C, was distributed onto autoclaved silicon molds (Microtissues 3D Petri Dish; Sigma-Aldrich), to generate 256-microwell casts (microwells: 300 µm diameter, 800 µm depth). Once solidified, the agarose casts were removed from the molds and each cast was placed inside the well of a 12-well cell culture plate with basal AdDMEM/F12 medium. Before use, the agarose casts were equilibrated for 1 h at 37 °C. The equilibration medium was removed and 256,000 cells, in a final cell suspension volume of 150 µL in Pneumacult™-Ex Plus medium, were seeded per cast. A resting period of 30 min was observed in order to allow the cells to sediment inside the microwells before adding 2 mL Pneumacult™-Ex Plus medium per well. Medium was changed twice a week. A total of 3 to 6 agarose casts were seeded with cells for each tissue sample.

### 2.4. Flow Cytometry Staining

Digested lung samples were washed in PBS-2% BSA-1 mM EDTA (FACS buffer), Fc-blocked (TrueStain, Biolegend) for 10 min, split into two, and stained with the relevant antibody or corresponding isotypes (0.5 µL/500,000 cells/200 µL) for 30 min at 4 °C. Immune cells were characterized with APC anti-human CD45 (H130, Biolegend). The endothelial cells and fibroblasts were analyzed with BB700 anti-human CD31 (WM59) and PE anti-human CD90 (5E10) (both from BD Biosciences, Bedford, MA, USA), and the epithelial cells with BV605 anti-human CD326 (EpCAM; EBA-1) (Biolegend). The controls received equivalent concentrations of isotype-matched IgG. The samples were washed, stained with DRAQ7 (Biolegend) and analyzed using an Attune NxT (Thermo Fisher Scientific, Rockford, IL, USA) analyzer.

### 2.5. Histology and Immunofluorescence

Immunofluorescence was performed on formalin-fixed, paraffin-embedded lung PDS. After fixation in paraformaldehyde 4% for 24 h, PDS cultured in agarose casts were covered with hot liquid bactoagar 2%. Once the casts were cooled down, they were dehydrated, embedded in paraffin and cut into 5 µm slices and put on slides. The sections were dewaxed, rehydrated, and subject to antigen retrieval via autoclave incubation in citrate buffer (0.01 M, pH 6.0) for CK7 and Ki67 staining, in Tris-EDTA buffer (0.01 M, pH 9.0) for CK5/6 and p63 staining, and in Tris-EGTA-buffer (0.01 M, pH 9.0) for TTF1 staining. For the intra-nuclear staining (TTF1, p63 and Ki67), the sections were permeabilized in PBS-0.5% Triton X100 at RT°C for 20 min. The slides were washed in PBS-1% BSA and incubated with primary antibodies (listed in the table below) at RT °C for two hours or at 4 °C overnight. After washing, the slides were incubated with secondary fluorochrome-coupled antibodies at room temperature for 1 h. Cell nuclei were counter-stained with Hoechst 33342 included in the mounting medium (ProTaqs Mount Fluor Anti-Fading, Biosystems). The antibodies are listed in Appendix A.

### 2.6. Cell Proliferation and Cell Apoptosis

Cell proliferation was estimated by immunolabeling cell nuclei with the Ki67 antibody (Cell signaling inc., Danvers, MA, USA) as detailed in the “Histology and immunofluorescence’ Section. The proportion of proliferative cells was calculated as the number of Ki67-positive cells over the total (Hoechst-positive) cell number. Cell apoptosis was analyzed using the TUNEL assay (In Situ Cell Death Detection Kit, TMR red, Sigma-Aldrich Chemie GMBH) following the manufacturer’s instructions, and the proportion of apoptotic proliferating cells was calculated as the percentage of TUNEL and Hoechst-positive cells (Zeiss Apotome, 20× magnification, Axiovision 4.6).

### 2.7. RNA Preparation and RT-qPCR

RNA was isolated from lung PDS cultured in 2 agarose casts (512 spheroids) for each condition. PDS were recovered after centrifugation of the inverted agarose casts for 5 min at 100G. Total RNA was extracted by using InviTrap Spin Universal RNA mini kit (Invitek Molecular GmbH, Berlin, Germany). cDNA was synthesized from 0.5 µg of total RNA using Reverse transcriptase (Invitrogen, Life Technologies). Quantitative PCR from 50 ng was performed in triplicate using Takyon™ No Rox SYBR Master Mix dTTP Blue (Eurogentec, Seraing, Belgique) and indicated primers. Gene expression was quantified using the DDCt method and normalized with three housekeeping genes (RPLP0, HPRT1, and GusB) using the primers listed in Appendix A.

### 2.8. Drug Sensitivity of PDS

After 3 days of culture in agarose casts, the PDS were transferred to a Cell-Repellent Surface 96-well microplate (one spheroid per well; Greiner Bio-One GmbH, Kremsmünster, Austria) and incubated with cisplatin (Cisplatine TEVA 100 mg/100 mL; Teva Pharmaceutical Industries, Petah Tikva, Israel)/etoposide (Selleck Chemicals, Houston, TX, USA) or cisplatin/paclitaxel (Selleck Chemicals) combinations for 3 h. The cisplatin/etoposide and the cisplatin/paclitaxel combinations were made by mixing cisplatin and etoposide or paclitaxel in a 1:0.675 and 1:0.8 ratio, respectively. The spheroids were treated with seven different concentrations of each combination ranging (two-fold serial dilutions) from 6.25/4.2 µM to 400/270 µM for cisplatin/etoposide, and from 6.25/5 µM to 400/320 µM for cisplatin/paclitaxel. The spheroids were cultured for an additional 3, 10 or 18 days after washed and their growth monitored by measuring their diameter. The PDS were imaged using BioTek Cytation 5 with corresponding software (Gen5, version 3.04) at the adjusted settings. PDS viability was assessed using the ATPlite™ assay (ATPlite™ 1stSTEP, PerkinElmer AG, Waltham, MA, USA) as described by Gendre DAJ et al. [27]. Graph Pad Prism 7 software (GraphPad Software, Inc., La Jolla, CA, USA) was used to analyze the data.

### 2.9. Gene Expression in TCGA

The mRNA levels of CADM1, KRT16, MMP1 and NAPSA in lung adenocarcinoma (ADC) were identified from the TIMER2.0 database (http://timer.cistrome.org/ (accessed on 22 august 2022)), which contains 10,897 tumor samples derived from 32 cancer types [27]. The diffexp module of TIMER2.0 was used to study the differential expression between tumor and adjacent normal tissues for these genes across the entire The Cancer Genome Atlas (TCGA) database of lung adenocarcinoma.

### 2.10. Statistical Analysis

Cell subset counts and statistics were analyzed using FlowJo (v.10.4.2) and Prism (v7.02) software via Mann–Whitney (2 groups) or Kruskal–Wallis (3 groups) tests. The data are expressed as mean ± SD (standard deviation) unless stated otherwise. Statistical differences between pairs of groups were determined using the unpaired Mann–Whitney U test. Differences were considered significant at *p* values < 0.05. Gene expression levels were presented using box plots and statistical significance was determined using the Wilcoxon test.

## 3. Results

### 3.1. Preparation and Characterization of Lung Tissues Samples

In order to establish patient-derived spheroids, surgically resected tissues were collected from lung carcinoma patients undergoing surgery procedures at the University Hospitals of Geneva. The samples included tumor lung tissue and non-malignant adjacent lung tissues (patient characteristics can be found in the Appendix A). The characterization of samples showed that the majority were obtained from patients diagnosed with lung adenocarcinoma (ADC, 60%), followed by squamous cell carcinoma (SqCC, 33%). The remaining samples included one sample obtained from an adenoid cystic carcinoma (Adenoid, 3%) and one sample from a lung carcinoid (Carcinoid, 3%). Most tumor samples originated from stage I (33%) and stage II (30%) tumors, with 17% and 20% of the tumor samples coming from stage III and stage IV resected tumors, respectively.

To establish spheroid cultures, lung tissues were first dissociated via enzymatic digestion to get individual cells or cell clusters. The number of cells obtained after enzymatic digestion was 11,806 cells/mg (6273–20,324; *n* = 25) for all lung sample types. Enzymatic digestion of adenocarcinoma samples resulted in a smaller number of cells compared to squamous cell carcinoma (Figure 1A right panel; *p* = 0.0448), where samples showed a positive correlation between the weight of the tissue and the number of cells obtained after digestion (Figure 1B). Cell subpopulations for non-malignant (henceforth referred to as “normal”) and lung tumor samples were determined using multiparameter flow cytometry (Gating strategy Appendix A and Materials and Methods for details). The data were analyzed in multiple ways to identify the percentage of live cells (DRAQ-negative), percentage of immune cells (CD45-positive), percentage of endothelial cells (CD45-negative, CD31-positive), and percentage of fibroblasts (CD45-negative, CD90-positive). The percentage of epithelial cells (CD45-negative, CD31-negative, CD90-negative) expressing the epithelial marker EpCAM was also determined. The viability rate of the lung cells after tissue digestion was high with 89.1% (85.6–92.8; *n* = 7) and 90.3% (79.7–95.2; *n* = 21) of viable DRAQ7^Neg^ cells for the normal and the tumor lung tissues, respectively (Appendix A). Most cells identified in the digested cell suspension were immune (CD45-positive cells), with a higher level of CD45^Pos^ cells in normal tissue (72.2%; 68.3–83.5; *n* = 7) compared to tumor tissue (59.6%; 31–72; *n* = 21; *p* = 0.0428) (Figure 1C; Appendix A). These values are consistent with previously reported proportions [28]. When comparing matched lung tissues (tumor and normal adjacent tissues from the same patient), we observed no significant differences in the ratio of immune cells. These data suggested that either the difference observed between tumor and normal lung tissues from all samples is associated with a patient-related difference rather than a tumor-related difference, or the number of matched samples used was too low to determine a statistical difference. Most cells identified in the non-immune CD45^Neg^ cell population were the epithelial cells CD45^Neg^-CD31^Neg^-CD90^Neg^-EpCAM^Pos^ and CD45^Neg^-CD31^Neg^-CD90^Neg^-EpCAM^Neg^ (other epithelial cells). The epithelial EpCAM^Pos^ cells represented 13–19% of the non-immune cells (Figure 1D, Appendix A). The remaining non-immune population was composed of CD90^Pos^-fibroblasts (12–13%) and CD31^Pos^-endothelial cells. The proportion of CD31^Pos^-endothelial cells was lower in tumor tissue (2.2%) compared with normal tissue (13.9%; *p* = 0.0313).

In summary, the enzymatic digestion of normal and tumor lung tissues allowed us to obtain viable cells for in vitro cell culture. We found that digested tissues contained subpopulations of different cell types (epithelial normal and tumor cells, fibroblasts, endothelial cells, and immune cells) and that CD45^Pos^ immune cells and CD45^Neg^-CD31^Neg^-CD90^Neg^ epithelial cells were the most prevalent cell types. Given that only 18 and 27% of all viable cells were lung normal and lung tumor epithelial cells, respectively, we selected for and expanded these cells as monolayer cultures prior to growing cells in spheroids.

### 3.2. Establishment of an Optimized Culture Method to Select for Primary Lung Normal and Tumor Epithelial Cells

We compared three different serum-free media designed for the expansion of epithelial cells to identify the optimal medium for the subsequent growth of primary lung normal and tumor epithelial cells. These included the culture media published by Kim M et al. [17] (noted C2), and two other media from STEMCELL technologies: Pneumacult™-Ex (PnEx) and Pneumacult™-Ex Plus (PnExP) media. The cells were seeded at different cell densities (Figure 2), and the rate of cell growth was estimated as the number of days to obtain cell confluency. The first epithelial colonies were observed at a median time of 6 (3–7) days after seeding (*n* = 61). The adherent cells progressed from sparse to confluent and were detached and reseeded at a median time of 14 (12–20) days after seeding (*n* = 55). No differences were observed by culturing the cells with the three media (Figure 2A).

Cell expansion in monolayer culture was successfully achieved for 71% and 70% of normal and tumor samples, respectively (12/17 from normal tissues and 14/20 from tumor tissues). As shown in Figure 2A, we determined a negative correlation between the cell density and the time of the first passage of the adherent cells (r = −0.3663, *p* = 0.0059). A cell density of 60,000–150,000 cells/cm^2^ led to an optimal cell growth with a median time of 12 (9–14; *n* = 15) days (Figure 2B). This culture condition is appropriate for an ex vivo model able to deliver results within a clinical time constraint of 3 weeks. Next, we compared the proliferation and apoptosis of normal and tumor tissue cells under the different culture media using Ki67 and TdT-mediated dUTP nick end labeling (TUNEL) staining (see methods), respectively (Figure 2C–E). Under all three conditions, the monolayers had similar dynamics, with about 10% of cells actively dividing (Figure 2D) and only about 1% of cells undergoing apoptosis (Figure 2E). Thus, no statistically significant difference was found in cell growth or viability between the three culture media tested. We chose the PnExP medium for subsequent in vitro culture of normal and tumor lung cells because it had previously been reported to sustain proficient growth and expansion of normal human bronchial epithelial (NHBE) [29]. Cell growth was successfully achieved in over 90% of samples using the PnExP medium, and led to a 20-fold increase in the cell population after 14 days of culture. We used multiparameter flow cytometry again to identify the cell subpopulations at the first passage and determine whether this culture step (monolayer culture in PnExP medium) allows for the enrichment of epithelial cells (Figure 3).

As expected, the in vitro culture of lung cells in PnExP medium favored a CD45^Neg^-CD31^Neg^-CD90^Neg^-epithelial cell subpopulation (Figure 3; Appendix A), which accounted for 88% (tumor) to 97% (normal) of the viable cells. Of these epithelial cells, 75% to 87% were EpCAM^Pos^ (Figure 3; Appendix A). Under monolayer culture conditions, there was a drastic reduction in immune (CD45^Pos^) cells in both normal and tumor fractions (0.7% to 1.6% of the viable cells) (Appendix A), and endothelial cells (CD31^Pos^) and fibroblasts (CD90^Pos^) made up less than 2.5% of the non-immune (CD45^Neg^) cultured cells.

### 3.3. Establishment of an Optimized Culture Method to Obtain Normal and Tumor Lung Spheroids

After expanding epithelial lung cells in the monolayer culture, we established 3D spheroids using a 256-well precast agarose mold (1000 cells/well) with PnExP medium. The cells first settled at the bottom of each micro-well and within 24 h of seeding, the cells aggregated to form PDS (Appendix A) measuring about 230 µm in diameter (210–250, *n* = 4). Consistent with what has been described previously [30], we observed a phenomenon of compaction from day 1 to day 3 following cell seeding (Appendix A), and a phenotype (loose or tight PDS, regular or irregular shape) that was dependent on the patient. The PDS could then be maintained in culture for two weeks (Appendix A). The expression of tumor lung cancer markers was assessed using immunofluorescence to confirm the lung phenotype of PDS. The PDS were stained for cytokeratin 7 (CK7), a member of the keratin family, and thyroid transcription factor 1 (TTF-1), both of which are normally expressed in type II pneumocytes and club cells in the lung. Two additional markers normally expressed in basal cells of the respiratory epithelium, cytokeratin 5 (CK5/6), another member of the keratin family, and tumor protein 63 (p63), were also used. As opposed to SqCC, which normally expresses CK5/6 and p63, NSCLC adenocarcinoma primarily expresses CK7 and TTF1 [31]. As expected, the majority of cells in the adenocarcinoma PDS expressed CK7 and TTF1 (Figure 4), whereas most cells in squamous cell carcinoma PDS expressed CK5/6 and p63 (Figure 4).

Despite the fact that the expression of lung cancer markers identified the cells in the PDS as epithelial lung cells, this characterization is not sufficient to conclude whether PDS cells are normal cells or tumor cells. Previous studies have supported the notion that pulmonary normal cells outgrow tumor cells and make up the majority of the cell population in cell cultures established from lung tumors [18,24]. To address this, we used the TIMER2.0 web server to analyze The Cancer Genome Atlas database and screen for genes differentially expressed between human lung adenocarcinoma and normal tissues. We selected four markers that were differentially expressed in normal and tumor human lung tissues: CADM1, KRT16, MMP1, and NAPSA (Figure 5).

RT-qPCR was used to analyze the expression of these markers on paired spheroids established from tumor and adjacent non-malignant lung samples of six patients (Figure 5). Of these six patients, five had lung adenocarcinoma (black symbols; 19LuCa04, 19LuCa05, 19LuCa06, 19LuCa08, 19LuCa12) and one patient had an adenoid cystic carcinoma (pink symbols; 19LuCa03) of the lung. The differential expression of the genes CADM1, KRT16 and MMP1 between the tumor and the normal PDS was similar in the patient with adenoid cystic carcinoma (19LuCa03) and in two patients with lung adenocarcinoma (black solid symbols; 19LuCa04, and 19LuCa12) to the TCGA lung adenocarcinoma analyzed via the Gen-DE module of the TIMER 2.0 web server (Figure 5). These results suggest that tumor cells made up most of the cells in the tumor PDS established from these three patients. As observed in the TCGA lung adenocarcinoma analysis, the paired PDS established from the other three patients with adenocarcinoma (empty black symbols; 19LuCa05, 19LuCa06, and 19LuCa08) revealed a lower expression of KRT16 between tumor and normal tissues. In this tumor PDS, the differential expression of the CADM1 and the MMP1 genes was reversed compared to that observed across the TCGA lung adenocarcinoma analysis. Moreover, the TP53 mutation (pVal157Gly) was initially identified in the tumor tissue sample taken from the adenocarcinoma patient 19LuCa06. We looked for this mutation in the tumor tissue, and in the tumor cells after enzymatic digestion and after culture in PDS (passage 1). While the TP53 mutation was detected in the tumor tissue and the tumor cells after enzymatic digestion, the tumor PDS did not harbor this mutation. This suggests that the PDS established from the 19LuCa06 tumor tissue was mainly composed of non-malignant cells.

To determine the viability of the cells in the PDS, cell proliferation and apoptosis were quantified (Figure 6). This revealed that there were only a few proliferating cells (green staining) and very few apoptotic cells (red staining) in the tumor PDS. The majority of the isolated cells located around the PDS were TUNEL-positive, suggesting that these are dead cells disaggregated from the PDS.

In summary, we were able to establish viable lung patient-derived spheroids (PDS) from normal and tumor lung tissues that were maintained in culture for two weeks. As published previously [18,24], our data from differential gene expression in paired PDS supported that around half of PDS established from lung tumors were mainly composed of tumor cells, while the other half was overgrown by normal pulmonary cells.

### 3.4. Study of the Effect of Standard Chemotherapeutic Treatments

To provide proof of principle that the PDS model used in this study is applicable for drug screening, we next assessed the efficacy of two chemotherapy combinations, cisplatin/etoposide or cisplatin/paclitaxel, on the PDS from the tumor of the patient with adenoid cystic carcinoma (19LuCa03). Three days after seeding, 19LuCa03 PDS were treated for 3 h with increasing concentrations of either the cisplatin/etoposide combination, the cisplatin/paclitaxel combination, or vehicle, and cultured for an additional 17 to 18 days after the drugs were washed out (Figure 7). Drug sensitivity of 19LuCa03 PDS was analyzed by quantifying PDS growth and viability (via intracellular ATP), as previously described by our group [30]. We observed a slight growth of control PDS (treated with vehicle) from day 3 up to day 21 in culture (Figure 7, left panels, solid black lines). In contrast, PDS exhibited a dose-dependent inhibition of growth (left panels) and viability (right panels) under both treatments (Figure 7). The highest doses of cisplatin/etoposide (400/270 µM) and cisplatin/paclitaxel (400/320 µM) caused a reduction in the PDS’s diameter of 39% and 24%, respectively (Figure 7, left panels), and a reduction in their viability of 100% and 71%, respectively (Figure 7, right panels), 17 to 18 days after treatment (Day 21 and Day 22 of culture).

These findings suggest that the cisplatin/etoposide combination inhibits the growth of 19LuCa03 tumor cells more efficiently than the cisplatin/paclitaxel combination. The disparity between the IC50 values of the two therapies, at 84/57 µM for cisplatin/etoposide and 103/82 µM for cisplatin/paclitaxel, emphasizes these findings. The efficacy of the carboplatin/pemetrexed combination was also assessed on PDS from the tumor of the patient with lung adenocarcinoma (19LuCa04; Appendix A).

In summary, the above findings show that PDS formed from adenoid cystic carcinoma was successfully used to evaluate the effectiveness of two chemotherapy treatments. Using the PDS model, we were able to determine that the cisplatin/etoposide combination was more effective on this patient’s cancer cells than the cisplatin/paclitaxel combination.

## 4. Discussion

Lung cancer high heterogeneity represents a significant challenge for the development of effective treatments. Precision oncology has become an attractive approach because it might enable oncologists to offer a specific treatment based on a molecular tumor identity card. However, this approach has proven to be successful only for specific cancer types. Furthermore, well-established predictive factors for response to treatment are lacking. The development of a model capable of testing individual patient response to different treatment options would thus greatly benefit clinical settings. Different 3D tumor models like PDS and PDOs have demonstrated relevance as culture models and hold promise for assessing personalized anti-cancer drug treatments. In this study, we successfully established PDS from tumor and non-malignant adjacent tissues of four subtypes of lung cancer: adenocarcinoma, squamous cell carcinoma, adenoid cystic carcinoma, and lung carcinoid. These subtypes cover over 85% of lung cancer cases. The PDS model was chosen due to its simplicity and reproducibility without the need for additional extracellular matrix, unlike PDOs.

The PDS model was generated by first culturing patient lung epithelial cells as adherent monolayers to remove immune cells and enrich with epithelial lung cells. This step facilitated the selection and expansion of epithelial cells, resulting in a 20-fold increase in cell number after two weeks of culture. It is worth noting that lung epithelial cells (both tumor and normal) could not be cultured long term in monolayer conditions, as previously reported [32]. After the fourth passage, their growth rate slowed, and the cells underwent senescence. Nevertheless, this step helped to obtain epithelial cells in sufficient numbers to establish an adequate number of PDS. The PDS could be used to evaluate the dose–response curves of two chemotherapeutic treatments and characterize the cells within the PDS, which maintained the expression patterns of subtype-specific markers. The genomic analysis of PDS established from six patients, including the expression level of four selected genes and the mutation profile, suggested that 50% of the PDS maintained a tumoral phenotype. Previous studies have reported that remnant normal cells in lung carcinoma samples tend to overgrow tumor cells [18,24]. Selecting tumor cells at an early step of the PDS establishment process holds promise in improving the quality and relevance of the tumor model. Strategies that allow for early cell selection, such as using cytologic quality evaluation and specific culture conditions, may also increase the rate of establishing tumor PDS. For instance, pure lung tumoroids have been successfully generated from primary lung tumors with mutant p53 after culture in the presence of Nutlin-3a. Nutlin-3a, a small molecule that inhibits the p53/MDM2 interaction, inducing autophagy and apoptosis in wild-type cells but not in p53 mutant cells, allows for the enrichment of tumor cells in culture [18]. Additionally, using a minimal medium that does not support the growth of non-cancerous cells can further enhance the success rate of establishing tumor PDS [17].

Our methodology is simple and both time and cost effective compared to other models like PDOs and patient-derived xenografts (PDX) models. PDX models grow too slowly to be used in a clinically relevant manner for drug screening. Instead, our model enables a quick and relevant assessment of chemotherapeutic treatments, enabling the identification of the most effective treatment for individual patients. Importantly, the use of PDS also aligns with the principles of the Three Rs (Replacement, Reduction, Refinement) of animal research [33], as it reduces the reliance on animal models and enhances clinical relevance.

A possible limitation of our PDS model is the absence of the tumor microenvironment, which can significantly influence cancer cell behavior [34,35]. A co-culture system that incorporates autologous fibroblasts and immune cells alongside lung tumor cells in 3D spheroid culture would help recapitulate the complexity of the tumor microenvironment, increasing the model’s predictive power for treatment outcomes. This would allow for an assessment of the efficacy of immune checkpoint inhibitors alone or in combination with chemotherapy or targeted therapy. The tumor microenvironment, and particularly cancer-associated fibroblasts, is known to play a crucial role in inducing drug resistance [36,37]. The inclusion of autologous cancer-associated fibroblasts in our heterotypic PDS composed of tumor lung cells offers significant potential to investigate and understand drug resistance mechanisms. To enhance the relevance of our model, we have analyzed the different cell populations obtained after tissue digestion. This characterization highlights the potential of our model for future work that involves the co-culture of heterotypic spheroids composed of these different cell populations. Importantly, although the pre-culture stage in adherent conditions may be perceived as a limitation due to the cell selection it induces, it becomes a pivotal asset for developing complex spheroids. This approach allows for cell type selection and facilitates the establishment of heterogeneous spheroids. Harnessing this approach can help address the concern of tumor microenvironment exclusion.

## 5. Conclusions

In conclusion, our PDS model represents a promising tool for screening new anti-cancer drugs and identifying personalized therapies and holds potential for future research involving the co-culture of heterotypic spheroids. PDS establishment from non-malignant adjacent lung tissues can also be utilized to estimate drug toxicity on normal lung cells. Ultimately, a more relevant and patient-specific in vitro system can help streamline drug discovery and accelerate the translation of promising therapies into clinical practice. In the future, we plan to conduct a clinical study in close collaboration with oncologists, where PDS will be established from biopsy specimens of lung carcinoma patients selected for therapy (chemotherapy or targeted therapy). The tumor sensitivity to prescribed anti-cancer treatments will be assessed using PDS and the results will be subsequently compared to the response of the tumor in situ.

## Figures and Tables

**Figure 1 cancers-15-05576-f001:**
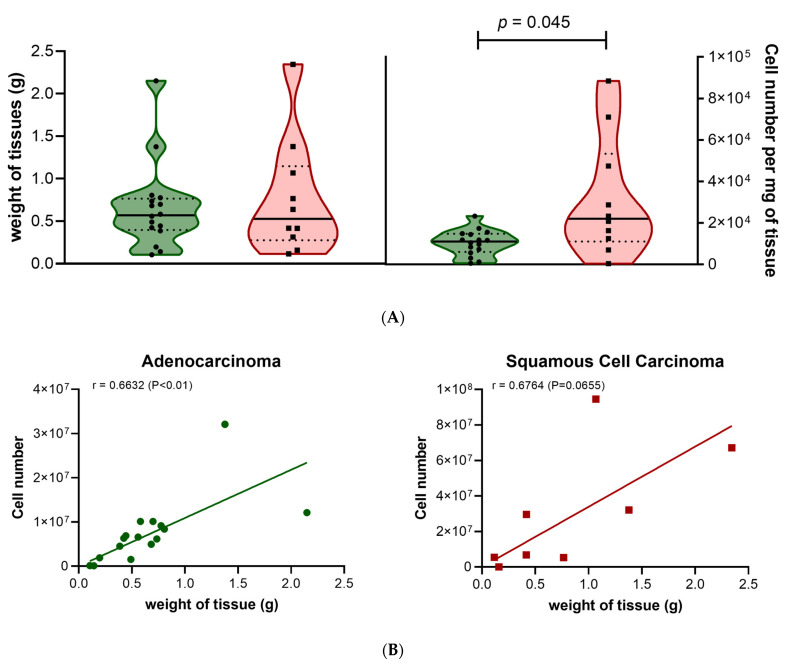
Characterization of cells from enzymatic digestion of lung tumor samples. (**A**) Comparison of resected lung adenocarcinoma (ADC, green) and squamous cell carcinoma (SqCC, red) weights (left) and quantity of cells obtained per milligram after enzymatic digestion (right). (**B**) correlation between weight of tissue and number of cells obtained after enzymatic digestion of lung adenocarcinoma (left, green) and squamous cell carcinoma (right, red). Data are presented as the median ± quartiles. (**C**) Comparison of immune (CD45Pos) cell percentage among viable cells in human lung normal (black) and tumor (white) tissues. (**D**) Comparison of cell subpopulations among non-immune cells (percentage of CD45Neg cells) in normal (left) and tumor (right) tissues using multiparameter flow cytometry. For (**A**), *n* = 16 samples for ADC and n= 8 samples for SqCC. For (**B**,**C**), *n*=7 samples for Normal and *n* = 21 samples for Tumor.

**Figure 2 cancers-15-05576-f002:**
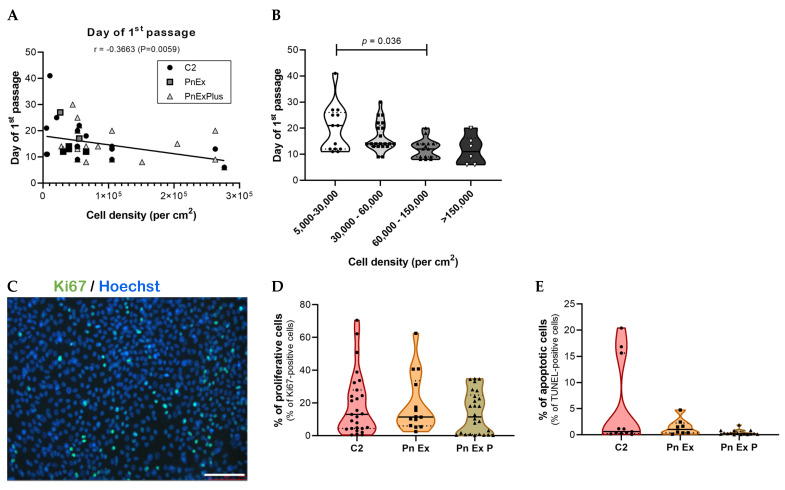
Cell growth and viability (proliferation and apoptosis) of human lung cells cultured in monolayer in three media. The cells dissociated from normal or tumor lung samples were seeded at different cell densities (from 5000 cells/cm^2^ to 300,000 cells/cm^2^) (**A**) A negative correlation was determined between the cell density and the time of the first passage. (**B**) The time for confluency was statistically reduced for a cell seeding density of 60,000–150,000 cells per cm^2^ compared with the other cell densities. Data are presented as the median ± quartiles of *n* = 24 samples. The cell proliferation and apoptosis levels were estimated by immunolabeling. (**C**) Representative immunofluorescence image of nuclei (Hoechst, blue) in human lung tumor cell monolayer shows Ki67-expressing cells (green) (scale bar = 500 µm). (**D**) Percentage of proliferative cells calculated as the number of Ki67-positive cells over total cell number Hoechst-positive cells were similar for the three media. (**E**) Percentage of apoptotic cells calculated as the number of TUNEL-positive cells over total cell number Hoechst-positive cells were similar for the three media. Data were presented as the median ± quartiles for samples from 4 patients.

**Figure 3 cancers-15-05576-f003:**
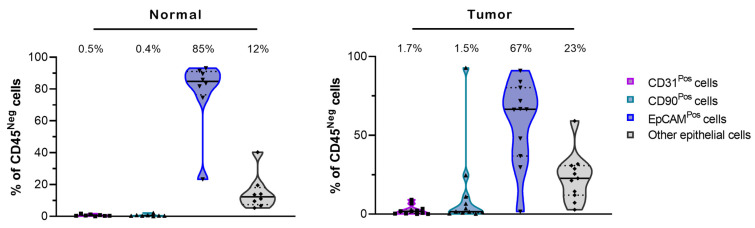
Characterization of cell subpopulations in monolayers cultured using PnExP medium. Comparison of cell subpopulations in non-immune cells (as percentage of CD45^Neg^ cells) in normal (**left**) and tumor (**right**) tissue cell samples cultured as monolayers in PnExP medium analyzed by multiparameter flow cytometry. Data are presented as the median ± quartiles of the cell proportion for *n* = 8 for normal samples and *n* = 11 for tumor samples.

**Figure 4 cancers-15-05576-f004:**
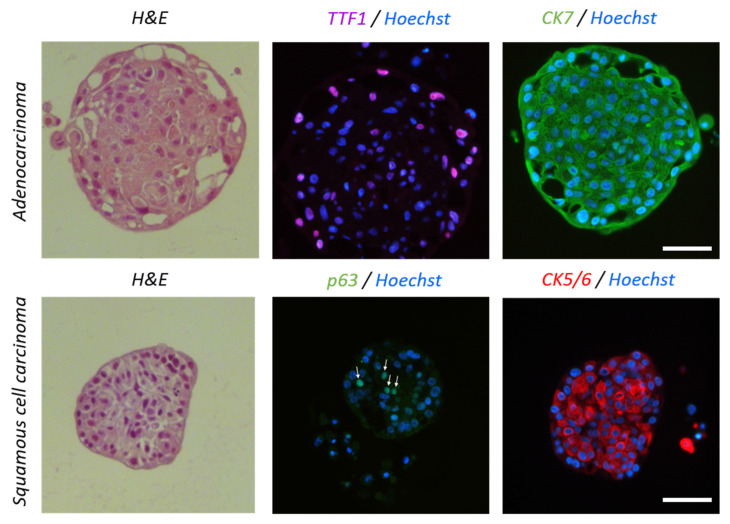
Expression of lung cancer markers in tumor PDS. Representative micrographs of adenocarcinoma (**top**) and squamous cell carcinoma (**bottom**) PDS stained with H&E, anti-TTF1, anti-CK7, anti-p63, and anti-CK5/6 antibody. Scale bars: 50 µm.

**Figure 5 cancers-15-05576-f005:**
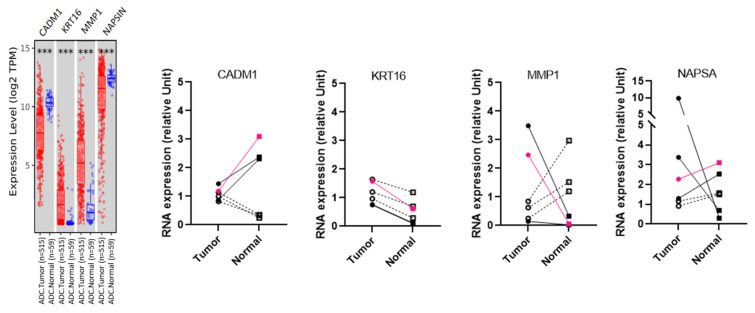
Gene expression analysis in lung adenocarcinoma. Differential expression of CADM1, KRT16, MMP1, and NAPSA genes between tumor (red) and adjacent normal (blue) tissues across TCGA lung adenocarcinoma (ADC) analyzed by the Gen-DE module of the TIMER 2.0 web server. Differential expression of genes in tumor (square) and adjacent normal (circle) tissues, analyzed by RT-qPCR, from 5 patients diagnosed with lung adenocarcinoma (black symbols) and 1 patient diagnosed with adenoid cystic carcinoma of the lung (pink symbols). Three paired PDS represented with filled symbols (circle and square) and straight lines had a differential expression of the four genes similar to the TCGA lung adenocarcinoma analyzed by the Gen-DE module of the TIMER 2.0 web server. The three other paired PDS, labeled with empty symbols and dotted lines, showed a different profile. *** *p* < 0.001.

**Figure 6 cancers-15-05576-f006:**
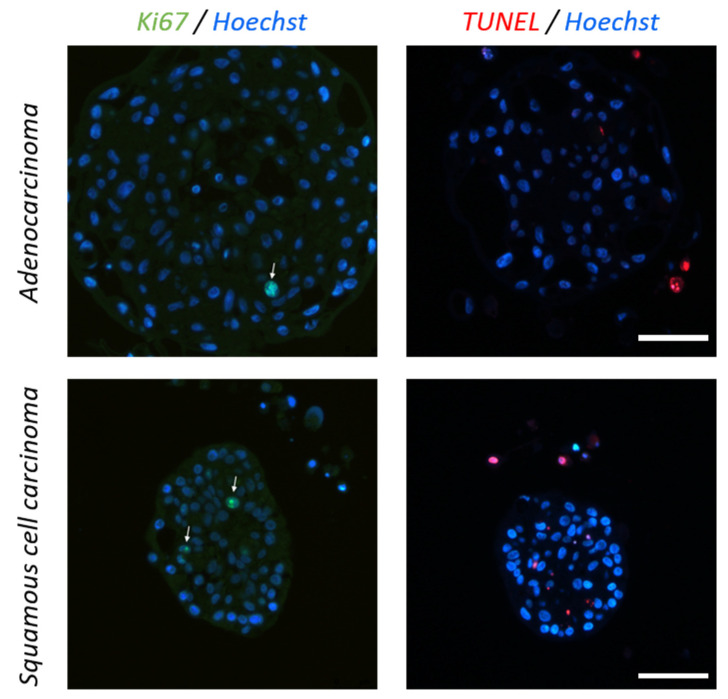
Lung tumor PDS proliferation and apoptosis. Representative immunofluorescence images of adenocarcinoma (top) and squamous cell carcinoma (bottom) PDS stained for proliferating cells (Ki67, green) and for dead cells (TUNEL staining, red). Scale bars: 50 µm.

**Figure 7 cancers-15-05576-f007:**
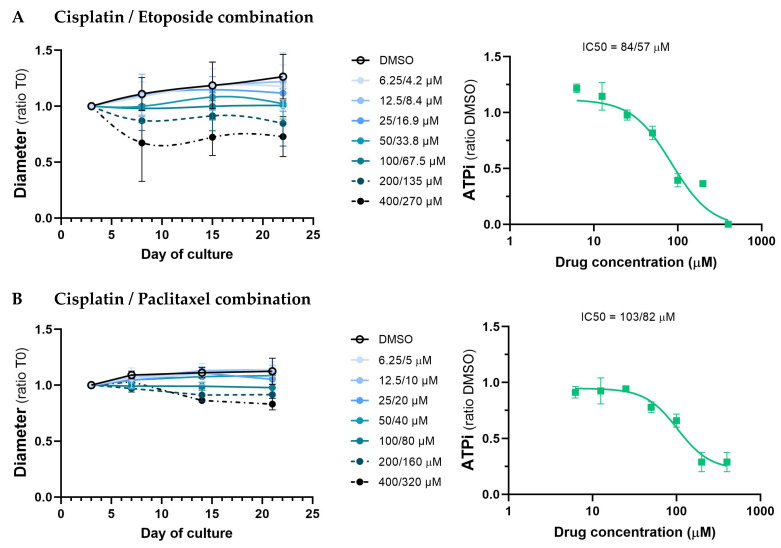
Dose-dependent effects of cisplatin/etoposide and cisplatin/paclitaxel treatments on adenoid cystic carcinoma PDS. For (**A**) Cisplatin/Etoposide treatment and (**B**) Cisplatin/Paclitaxel treatment, assessment of PDS growth (left panels) by quantification of PDS spheroid diameter and assessment of PDS viability (right panels) at day 14 after treatment using intracellular ATP (ATPi). The values of the x-axis of the ATPi panels (right) are Cisplatin concentrations. Data are presented as mean ± SEM of 3 to 9 PDS from a single experiment.

## Data Availability

No new data were created.

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
