# Peer review of "An Optimized Method to Culture Human Primary Lung Tumor Cell Spheroids"

_cancers, 2023, doi:10.3390/cancers15235576_

Round 1

Reviewer 1 Report

Comments and Suggestions for Authors

This work is exciting and well-prepared and I have only minor comments:

- The manuscript should have been prepared according to the author's guidelines.

- More details should be provided in all the tables in the methods section.

- Some references are old references and there are new versions of the manuscript. Please use the new papers instead of the old papers.

Comments on the Quality of English Language

Minor editing of English language required

Reviewer 2 Report

Comments and Suggestions for Authors

The manuscript presents a novel ex-vivo method based of patient derived tumor spheroids to test drug sensitivity. This may provide information on the best treatment for each patient within a few weeks. The authors perform an extensive characterization of the spheroids and their constituent cells. Further evidence would be required to support the use of this model to estimate drug sensitivity. 

1. The order of the references needs to be checked. For example, after reference 24 (presented in line 81), the next reference is number 29 (line 90). 

2. Line 105. It is not clear, whether the 256 spheroids were generated from each tumor. Please, clarify

3. Please, indicate the dilution of all the antibodies used in the study

4. Line 131. There is a typo in the units. 

5. Lines 161-162. Please, indicate the concentration of paclitaxel and etoposide used. 

6. PDS viability was determined with the ATP Lite kit. This kit measures total ATP. In this regard, changes may be due to alterations in the metabolism of the spheroid instead of changes in the cell viability. The authors should use an alternative method to support the conclusions on the PDS viability. 

7. Lines 259-261. It is not clear how the % of normal lung and tumoral lung cells was determined. Please, clarify

8. Figure 4. The labelling of the lower right panel is not correct (should be CK 5/6).

9. Figure 5. The meaning of the black filled and the empty squares and circles is not clear. Please, clarify in the figure legend. While in the figure legend, patients are divided in those with adenocarcinoma (black) and adenoid cystic carcinoma (pink), a third group seems to be present in the figure. 

10. Figure 7, ATPi panels. Since spheroids were treated with two different drugs, it is not clear how the values of the x-axis (concentration) are calculated.

11. The choice of a PDS derived from adenoid cystic carcinoma to perform drug sensitivity tests is not clear. This is, in fact, a rare type of tumor. In order to support the conclusions of the manuscript, similar tests need to be performed in the other types of PDS.

12. It would be very useful if the authors have access to clinical data on the chemotherapy scheme received by the patients and the clinical outcome and compare these data with the sensitivity/resistance exhibited by the corresponding PDS. 

Reviewer 3 Report

Comments and Suggestions for Authors

Véronique Serre-Beinier et al. reported an optimized method to culture human primary lung tumor cell spheroids. The topic was interesting and to some degree significant, and might arouse a certain impact in its field. The reported data was relatively convincing. Overall, the article fell within the scope of Cancers. The reviewer suggested a Minor Revision prior to a final acceptance. Please refer to the following comments:

1) Please revise the format of Reagents in the Methods Section, which was not the common style of biomedical research.

2) All tables should be accompanied with captions (Table 1 ……), including the details of RT-qPCR.

3) The resolution of figures was relatively low, especially Figure S1. Please substitute it with a high-res figure.

4) Instead of placing an asterisk (*) in Figure 1, 2, etc., it would be better to directly provide the p values.

5) It was suggested to add some personal opinion about the application of the developed model on multi-drug resistant cancers, in the Discussion Section.

6) More valuable information could be shown in the Conclusion Section, which was currently less than 80 words.

Round 2

Reviewer 2 Report

Comments and Suggestions for Authors

The authors have addressed all my concerns and delivered a significantly improved version.